# Single-defect spectroscopy in the shortwave infrared

Xiaojian Wu [1], Mijin Kim [1], Haoran Qu [1] & YuHuang Wang [1,2]

Chemical defects that fluoresce in the shortwave infrared open exciting opportunities in deep-penetration bioimaging, chemically specific sensing, and quantum technologies. However, the atomic size of defects and the high noise of infrared detectors have posed significant challenges to the studies of these unique emitters. Here we demonstrate high throughput single-defect spectroscopy in the shortwave infrared capable of quantitatively and spectrally resolving chemical defects at the single defect level. By cooling an InGaAs detector array down to −190 °C and implementing a nondestructive readout scheme, we are able to capture low light fluorescent events in the shortwave infrared with a signal-to-noise ratio improved by more than three orders-of-magnitude. As a demonstration, we show it is possible to resolve individual chemical defects in carbon nanotube semiconductors, simultaneously collecting a full spectrum for each defect within the entire field of view at the single defect limit.

[1] Department of Chemistry and Biochemistry, University of Maryland, 8051 Regent Drive, College Park, MD 20742, USA. [2] Maryland NanoCenter, University of Maryland, College Park, MD 20742, USA. Correspondence and requests for materials should be addressed to Y.H.W. (email: yhw@umd.edu)

Chemical defects are nearly ubiquitous in low-dimensional materials. However, their roles are largely unknown[1–4]. The challenges are primarily due to the small footprint of these defects and the extended lattice of the solid substrate, which make the task of finding a defect like searching for a needle in a haystack. Additionally, defects may occur in random clusters that are indistinguishable from each other or they may present in different atomic configurations, each of which modifies the host to a different degree[3,5,6,7]. The ability to study defects at the single defect limit will circumvent the limitations imposed by ensemble averaging and unravel the roles of defects on the chemical, optical, and electronic properties of a material[1,4].

The shortwave infrared (typically 900–1700 nm, but may also refer to 1–3 μm) is a spectral window that presents exciting opportunities for bioimaging[8,9], telecommunication[10], and quantum technologies[11,12]. For instance, optical-fiber-based quantum communications require single photon sources that emit in the telecom bands (1260–1625 nm)[10,13]. Color centers (emissive point defects in crystal lattices) that emit light in the shortwave IR are also important for non-invasive in vivo fluorescent imaging[8,14]. The deep penetration of shortwave IR light in living tissue[15], as well as the negligible tissue autofluorescence within this optical window[16], offers enhanced resolution and contrast[17,18]. Although spectroscopic studies of individual color centers in the visible range have been reported[19,20], single-defect spectroscopy in the shortwave IR is challenging due to the diffraction limit, which is approximately half of the wavelength of the emitted light, and indium gallium arsenide (InGaAs) detectors, which are noisy for low-light measurements[21,22]. Additionally, the throughput of existing methods is typically low because the sample is scanned point-by-point.

Here we show it is possible to perform high throughput single-defect spectroscopy in the shortwave IR to spectrally identify and quantitatively count chemical defects at the single-defect level. We show that nondestructive readout and cooling the InGaAs detector array to −190 °C (vs. −100 °C typical of liquid-N$_2$ cooled detectors) collectively improve the signal-to-noise ratio by more than three orders-of-magnitude. We have also designed a Si-based fiduciary marker that works in the shortwave IR to correct the stage drift at a resolution of 5 nm. As an illustration of this high-resolution high throughput single-defect spectroscopy method, we show it is possible to simultaneously resolve chemical defects in carbon nanotube semiconductors at the single-defect limit, collecting a full spectrum for each defect within the entire field of view.

## Results and discussion

**Counting chemical defects.** Our strategy for counting defects is based on quantitative analysis of the stochastic blinking of defect photoluminescence (PL). We hypothesize for an emitting site that contains only one defect, PL blinks on and off at a constant intensity step (Fig. 1a, b). If multiple defects are present at an emitting site, the PL time trajectory will have a variable number of intensity states and multiple step sizes depending on the number of defects present. For example, when three defects are blinking independently, they will collectively produce four PL intensity states and three step sizes (Fig. 1c, d). Although less likely, adjacent defects may blink in a correlated manner, such that two or more defects are turned on or off in synchrony. However, we can retrieve the defect locations in a manner equivalent to the super-resolution approach used in stochastic optical reconstruction microscopy[23]. By analyzing the two-state nature of fluorescence blinking from a single-defect emitting site in correlation with its super-resolved location, we can optically identify single defects. This approach, when combined with the hyperspectral

capabilities of our imaging system, makes it possible to study emissive defects at the single-defect level.

**Super-resolved hyperspectral imaging system.** To perform single-defect spectroscopy in the shortwave IR, we custom-built a super-resolved hyperspectral imaging system that allows the acquisition of PL spectra from all pixels in the camera frame simultaneously (Fig. 2a). To achieve this, a volume Bragg grating (VBG) is placed in front of the detector on the light path and an image stack is collected, one wavelength at a time, and reconstructed to provide the PL spectrum for each pixel over the entire field of view (see "Methods"). This grating can be bypassed so the system can be switched between the spectrum mode and the imaging mode. In imaging mode, the stochastic blinking of defect emissions is recorded and analyzed to super-resolve defect locations[21]. We can thus spectrally and spatially resolve the same defects at the single-defect level by simultaneous super-resolution[21] and hyperspectral imaging[24].

Additionally, we have developed a drift correction method that works in the shortwave IR. Although fiduciary markers, such as fluorescent beads immobilized on the sample substrate[25], have been used to correct stage drifts, they work only in the visible range. Our design makes use of silicon, which is transparent in the shortwave IR since its bandgap absorption cuts off at 1100 nm, to directly create shortwave IR markers on the imaging substrate. The markers used in this experiment are 5 μm diameter dots of shortwave IR-transparent windows, which are patterned at a pitch of 30 μm on a silicon wafer by photolithography, with Au covering the rest of the substrate (Fig. 2b and Supplementary Fig. 1). This design results in a photomask-like structure allowing the shortwave IR component of the microscope's halogen lamp to transmit through the patterned Si dots to create bright markers while the visible component is blocked by the silicon (Fig. 2b). The Au layer concomitantly acts as a mirror to enhance the collection efficiency of photons emitting from the sample while blocking the fluorescence from the underlying silicon substrate. To prevent PL quenching by Au, a ~50 nm thick polystyrene (PS) layer was spin-coated on top of the Au layer. With this substrate design the collection efficiency of the nanotube PL is ~5-times higher than when imaged on bare glass (Supplementary Fig. 2). This observed increase can be attributed to the reflective gold layer acting as a mirror to enhance both the absorption of excitation photons by the carbon nanotubes and the collection efficiency of emission photons by the objective, as well as the PS layer, which minimizes surface charges that would otherwise quench the PL. Figure 2c shows the markers simultaneously resolved along with single-walled carbon nanotubes (SWCNTs) by the InGaAs camera. We fit the intensity profile of each individual dot marker with a 2D Gaussian to locate its center (Supplementary Fig. 3c), and this process was repeated for all imaging frames in a time sequence, allowing us to plot the sample drift trajectory (Supplementary Fig. 3d, e). The standard deviation of an individual drift trajectory from the averaged one defines the precision of this drift correction, which we determined to be 5 nm (Supplementary Fig. 3f). We note that to achieve 5-nm drift correction precision, we ensured there were at least five markers in the field of view.

To detect extremely low light stemming from individual defects in the shortwave IR, we integrated a liquid nitrogen cooled InGaAs detector array (Cougar 640) that is optimized for −190 °C operation. Note that liquid-N$_2$ cooled InGaAs detectors are typically operated at temperatures no lower than −100 °C. Cooling down to nearly the true liquid-N$_2$ temperature suppresses the dark current to just 10 e$^-$ pixel$^{-1}$ s$^{-1}$, which is 500 times lower than achievable by InGaAs detectors cooled to −100 °C.

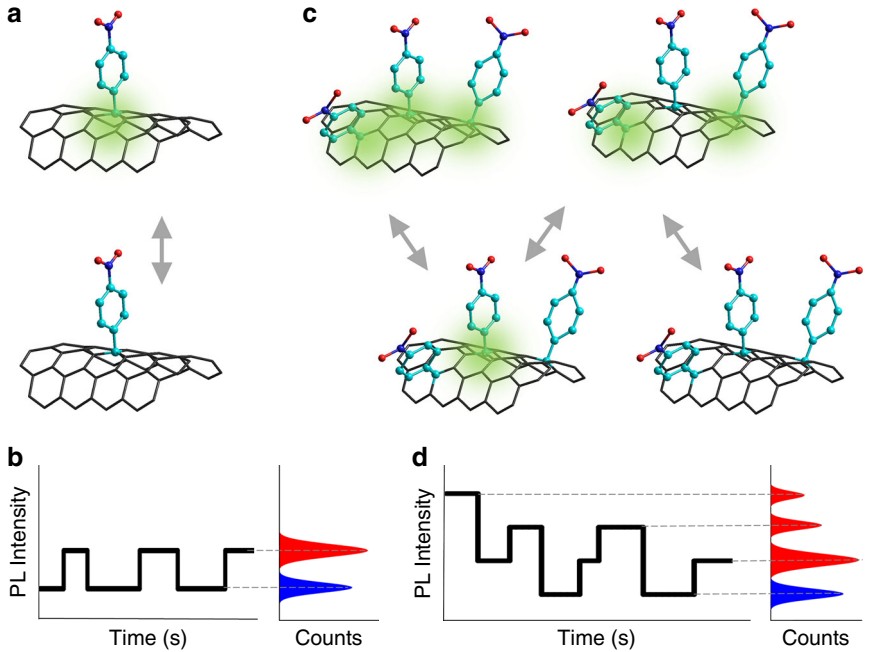

**Fig. 1** Counting defects based on photoluminescence (PL) blinking. **a** Schematic illustration of a single chemical defect that switches between PL on (green shade) and off states due to surface charges. The gray arrows represent reversible PL state switching. **b** The illustration of PL intensity time trajectory (black solid line) and histogram from a single-defect features two states, PL on and off. The red and blue bells in the histogram represent the distribution of PL intensity of on and off states respectively. Dashed lines are added as a guide to the eye. **c** For a cluster of defects, each defect randomly turns on and off. **d** The stochastic blinking of clustered defect PL is collectively shown as multiple intensity steps

Another enabling feature of our system is the implementation of a read-while-integrate (RWI) readout mode, which significantly reduces the read noise. Unlike the conventional integrate-then-read (ITR) mode, RWI is non-destructive, which eliminates the reset and read noise. Figure 2d compares the signals of the same pixel from an image sequence captured by the two modes to demonstrate the working principle of RWI. In the RWI mode, the signal is read without resetting the detector while the ITR mode resets after each signal readout. Since the pixel is reset to a slightly different value each time, this causes significant reset noise in ITR (Fig. 2d). This reset noise is effectively removed in the RWI mode since the reset is avoided. The noise in RWI can be further suppressed by linear regression to eliminate the read noise. For each image, we accumulate $n$ readouts, from which the slope of every pixel is calculated by

$$\alpha = \left[ \frac{n\sum(xy) - \sum x \sum y}{n\sum x^2 - (\sum x)^2} \right] \quad (1)$$

where $x$ is the readout count and $y$ the readout value. We then multiply the calculated slope $\alpha$ with the total number of accumulated readouts $n$ for each pixel (i.e., Pixel $= n\alpha$) to obtain the signal intensity. The effect of the RWI mode on reducing the image noise is dramatic, increasing the signal-to-noise ratio by more than 10 times (Fig. 2e). The clean background significantly improves the image contrast. Supplementary Figure 4a, b show PL images of the same SWCNT with the same integration time (1 s) captured in the ITR and RWI modes, respectively. The nanotube is clearly resolved in RWI but almost invisible in ITR. The much higher image contrast in RWI is due to a much lower noise level while maintaining the signal intensity, as shown in the intensity profile (Supplementary Fig. 4c).

The RWI mode also greatly reduces the background signal fluctuation over time in an image sequence. RWI is a continuous readout mode (Fig. 2d), which reads the accumulated signal at a preset time interval. In order to capture the PL time trajectory, we

recorded the luminescence image sequence at 2 frames per second and then computed the difference between two adjacent images in the readout sequence (which accumulate the signal from time zero up to the point of each readout) using MATLAB codes to obtain the time-lapse images at equal time intervals. Supplementary Fig. 4d shows a 60 s intensity trajectory of the same pixel taken using the RWI and ITR modes under the otherwise same conditions. RWI demonstrates a background fluctuation significantly lower than the ITR mode, providing the sensitivity required to probe the defect PL fluctuation in a time sequence.

**Characterization of organic color-centers**. As an illustration of this high sensitivity single-defect spectroscopy, we characterized organic color-centers which are synthetic defects chemically incorporated into SWCNTs by covalently bonding functional groups to the sidewalls[26–29]. The introduced chemical defect locally modulates the electronic structure of the SWCNT in such a way that mobile excitons can become trapped at the defect site where they emit brightly as single photons in the shortwave IR[1,4,11,12]. We excited the sample at an off-resonant wavelength at low power densities (below 64 W cm$^{-2}$) to avoid exciton-exciton annihilation effects[30]. We find that the defect emission is highly localized (Fig. 3a) and the defect emission shows stochastic PL intensity changes (Supplementary Movie 1). This PL blinking can be suppressed when the SWCNT is sandwiched between two layers of polystyrene, which form a charge-free environment[31], confirming surface charges as the cause of blinking. This intermittency in defect emission enables us to identify single defects by analyzing the PL intensity time trajectory from a defect emitting site. Figure 3c shows the PL intensity trajectories acquired over 10 min, revealing the presence of multiple intensity states for the emitting defect sites marked in Fig. 3a. We quantified the number of intensity states by analyzing the PL intensity distribution, presented as a histogram in Fig. 3d, which is bimodal for (i) and (ii), but trimodal in (iii). Although the defect emitting sites in all

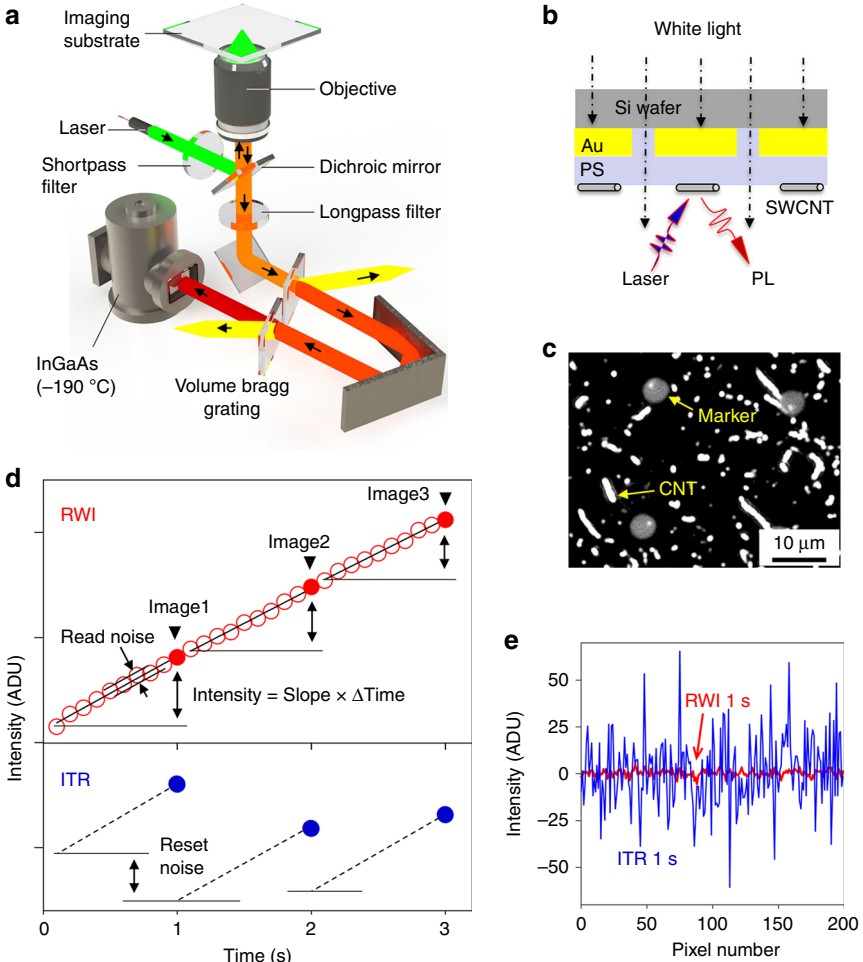

**Fig. 2** Single-defect spectroscopy set-up. **a** Schematic of the system, which integrates a volume Bragg grating for hyperspectral imaging and an InGaAs 2D array detector that operates at −190 °C to significantly suppress the dark current in the shortwave IR. **b** Schematic of the imaging substrate that incorporates Au-on-Si markers and polystyrene (PS) insulating layer for drift correction in the shortwave IR. **c** A PL image with both SWCNTs and the markers simultaneously resolved. Note that nanotubes with length smaller than the diffraction limit (~600 nm for a numerical aperture 0.85 objective collecting emission at 1000 nm) appear as white dots in the image. **d** The comparison of signals of the same pixel from an image sequence captured by the integrate-then-read (ITR, blue filled circles) and read-while-integrate (RWI, red empty circles) demonstrates the working principle of non-destructive RWI mode. The integration slopes for different images in ITR (dashed lines) do not have identical starting point, caused by reset noise. Unlike ITR, RWI does not reset the detector, thus eliminating the reset noise. For each image (red filled circles) in RWI, the noise can be further suppressed by linear regression of different data points to retrieve the integration slope (black solid line). **e** The comparison of background signals from images captured by the ITR (blue) and RWI (red). Under otherwise identical conditions, RWI demonstrates more than an order of magnitude lower noise level than ITR, with a standard deviation of merely 1.75 analog-digital unit (ADU) (vs. 18.5 ADU for ITR). Note that 0.168 ADU corresponds to one photon arriving at the detector

three cases appear as similar diffraction-limited spots, the PL intensity time trajectories are bimodal vs. trimodal, which are characteristic of one (vs. two) emitting centers in those sites. We note that the majority (>80%) of defect emitting sites observed in the current study show multiple intensity steps, indicating the prevalence of clustered defects.

To spatially resolve defects within a cluster, we retrieved the defect locations by fitting the differential images (between successive images before and after each PL blinking event), as we have previously shown to super-resolve ultrashort nanotubes[21], with a Gaussian approximation of the point spread function of our setup. Since each differential image represents the appearance or disappearance of the PL stemming from a defect within a cluster, we can compute and compare the locations of defects to determine the number of defects at the limit of the localization precision. Figure 3b shows the defect localizations for several emitting defect sites marked in Fig. 3a. Here, individual defect localization is displayed as blue spots with the

corresponding localization precision as dotted circles. Note that each localization has different localization precision, determined following a literature method[23,32], and the localization precision is as high as 15 nm in our experiments. The localization confirms the presence of only one single defect at emitting site (i) and two defects at (iii), which are consistent with their PL blinking behavior.

Unexpectedly, we identified two defects from emitting site (ii) even though the PL from this site shows two-state blinking. In addition, the PL time trajectory of (ii) shows blinking at step heights that are quantitatively correlated with two defects. This observation suggests the two defects are strongly coupled, manifesting as a two-state on and off mode just like a single defect. This correlation makes it possible to distinguish independent defects from coupled defects that are in very close proximity to each other. This technique suggests the possibility of quantitatively measuring the coupling effect of two or more closely spaced defects. We note that this coupling effect was

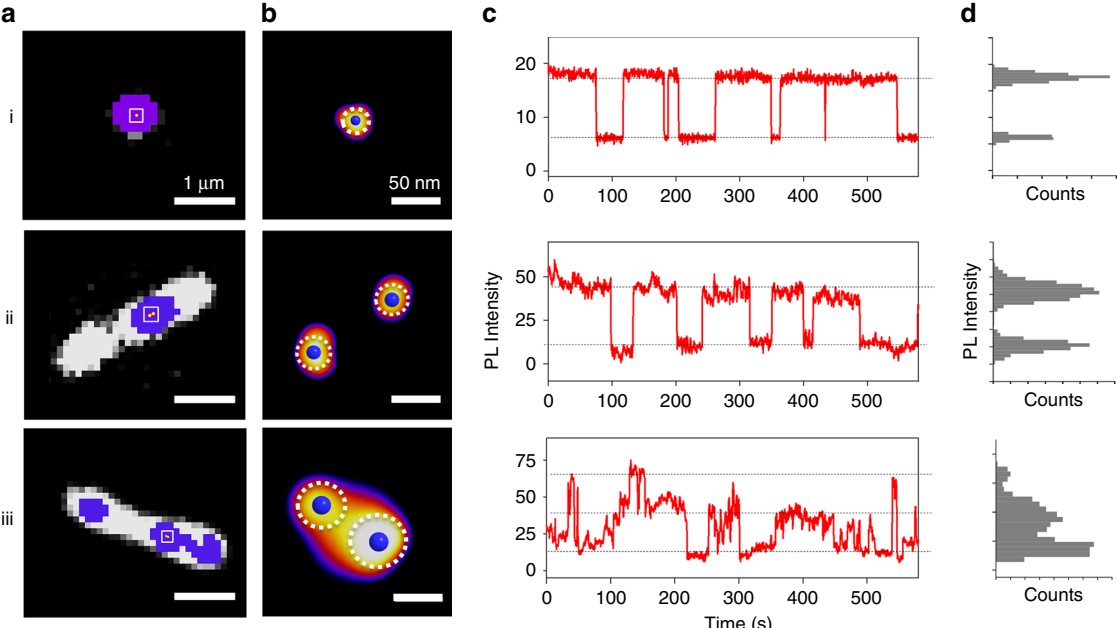

**Fig. 3** Super-localization and PL blinking of individual 4-nitroaryl defects. Note that the images on the same row are associated with the same defect sites. **a** The diffraction-limited PL images of the defect emission sites (blue) and intrinsic $E_{11}$ exciton emission (white) reveal the localized nature of the defects along the nanotube length. **b** Individual defects within those emitting sites marked in (**a** white square) are resolved with sub-pixel resolution. Here, individual defect localization is displayed as blue spots with the corresponding localization precision as dotted circles. **c** PL intensity trajectories (red solid line) from each defect emitting sites reveal stochastic blinking behavior and **d** the corresponding PL intensity histograms show bimodal (i) and (ii), and trimodal (iii) intensity states. Dashed lines are added as a guide to the eye

observed only occasionally (~5%) in the current study and will be further exploited in future experiments.

We further showed it is possible to address the outstanding challenge of unambiguously determining the chemical nature of defects at the single-defect level[5,6,33]. To demonstrate this possibility, we synthesized (6,5)-SWCNTs containing two types of defects: methoxyaryl (MeOAr-) and nitroaryl (NO$_2$Ar-) (Fig. 4). Figure 4a shows a PL image with the defect locations super-resolved (Fig. 4a). Note that the defects are located inhomogeneously along the nanotube length. We then plotted the PL spectra from each pixel along the tube length in a spatially and spectrally correlated map with the defect PL resolved (Fig. 4b). From this particular nanotube, we identified three defect emitting sites with overlapping defect emission spectra but distinct center wavelengths, spatially corresponding to the three super-resolved defect groups in Fig. 4a. By comparing these results with the spectra of isolated defects (Fig. 4c), we can unambiguously assign each defect as (i) nitroaryl defects; (ii) methoxyaryl defects; and (iii) both. We note that the two defects at site iii show a 5 ~ 6 meV redshift in emission energy compared to the corresponding defects at sites i and ii. This shift is relatively small compared to the energy difference between these two types of defects (~19 meV)[29] and was also observed in the $E_{11}$ emission (Supplementary Fig. 5). This behavior may arise from coupling between neighboring defects or the different local dielectric environment caused by inhomogeneous surfactant coverage on the nanotube surface[34] or the partial filling of water inside the SWCNT[35]. Further studies are warranted to understand and quantify these different effects.

In summary, we have demonstrated a super-resolved hyperspectral imaging technique capable of performing high-throughput single-defect spectroscopy in the shortwave infrared. Distinct from conventional single point spectroscopy, which requires point-by-point scanning, our method allows all 640 ×

512 pixels of an InGaAs detector array to simultaneously acquire a full PL spectrum to achieve hyperspectral function. Notably, by cooling the detector array to nearly the true liquid-nitrogen temperature (−190 vs. −100 °C) and implementing a non-destructive readout scheme, we reduced the dark current by nearly 500-fold and achieved another 10-times improvement in the signal-to-noise ratio due to the low-noise readout, collectively improving the signal-to-noise ratio by more than three orders-of-magnitude. Additionally, we have designed a fiduciary marker that works in the shortwave IR to achieve a correction precision as high as 5 nm. As an illustration of this super-resolved hyperspectral imaging technique, we resolved individual chemical defects, both spatially and spectrally, along SWCNT semiconductor hosts. We show it is possible to capture low light fluorescent events at the single emissive defects, allowing us to quantify the number of defects within a diffraction-limited spot and spatially resolve them with a resolution of 15 nm in the shortwave infrared with simultaneous hyperspectral capabilities to unambiguously determine their chemical identities. This super-resolved hyperspectral imaging technique may open possibilities to screen shortwave IR fluorophores, resolve functionalization patterns, capture single molecule reactions, and probe chemical defects with high throughput and rich spectral details at the single-defect limit.

## Methods

**The single-defect spectroscopy imaging system.** The system was built on a Nikon Eclipse U inverted microscope with IR optimized objectives, including a 100× objective (LCPLN100XIR, numerical aperture (NA) = 0.85, Olympus) and a 150x objective (UAPON150XOTIRF, NA = 1.45, Olympus). A continuous wave laser beam at 730 nm (Shanghai Dream Lasers Technology Co., Ltd.) or 561 nm (Jive$^{TM}$ Cobolt AB, Sweden) was shaped through a beam shaping module to produce a top hat profile, with the top flat part of the hat effectively overfilling the field of view on the sample surface. The laser was then reflected into the objective through a long pass dichroic mirror (875 nm edge, Semrock, USA) and focused to

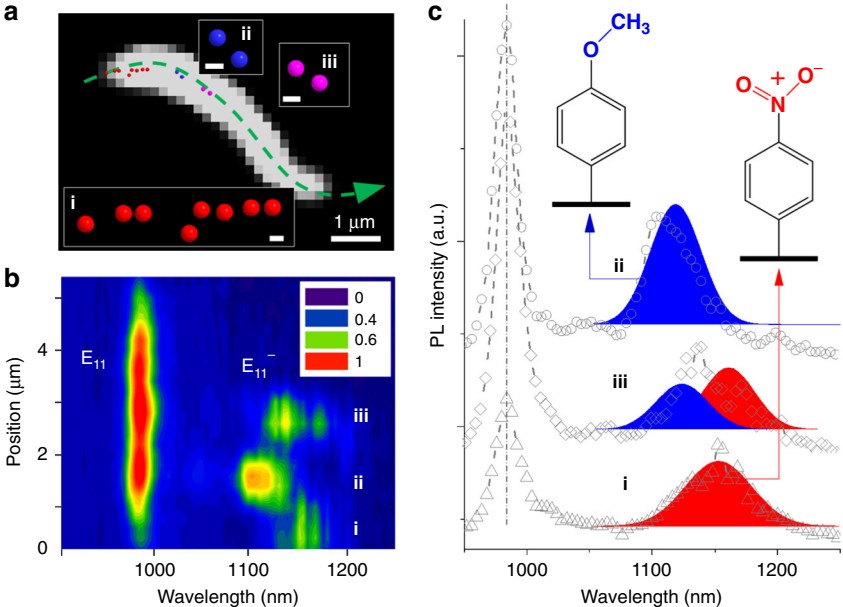

**Fig. 4** Resolving and identifying two types of defects along a nanotube. **a** The super-resolved defect locations superimposed on the $E_{11}$ emission of a (6,5)-SWCNT. Three major defect groups can be found. The insets are close looks of the resolved defect locations from these three defect groups. Red, blue and magenta represent 4-nitroaryl, 4-methoxyaryl and mix of the two defect types, respectively (scale bar 100 nm). **b** Spatially and spectrally correlated PL map along the nanotube centerline as indicated by the green dashed line in (**a**). The y-axis represents the physical positions on the centerline, starting from the left nanotube end. Three defect sites with emission wavelengths in the range of 1080 to 1200 nm can be identified, which spatially correspond to the three super-resolved defect groups in (**a**). **c** PL spectra from the three identified defect sites. Site i and ii are 4-nitroaryl and 4-methoxyaryl defects, respectively, and site iii contains both

create a uniform excitation field on the sample surface. Fluorescence emission from the sample was collected by the same objective and filtered through the long pass dichroic mirror to remove the photons produced by elastic laser scattering at the sample surface. The detector for the fluorescent signal was a Cougar-640 imaging camera (Xenics, Leuven, Belgium) with an InGaAs focal plane array with $640 \times 512$ pixels, cooled with liquid nitrogen to $-190\,°C$. To perform hyperspectral imaging, a volume Bragg grating (VBG, Photon Etc., Montreal, Canada) was situated between the long pass dichroic mirror and the liquid nitrogen cooled InGaAs detector array. The fluorescent signal was filtered by the VBG and only the diffracted light with a narrow bandwidth of 3.7 nm was directed to the detector to form a spectrally-filtered image. A hyperspectral cube containing a stack of images at spectral intervals of 4 nm was collected by rotating the VBG to continuously tune the diffracted wavelength. To record the defect emission blinking in the full field of view, the VBG was bypassed, where instead a 1100 nm long pass filter (Thorlabs, FELH1100) was placed between the long pass dichroic mirror and the InGaAs detector array to block the fluorescent signal from the semiconductor nanotube itself (~980 nm for (6,5)-SWCNTs) and image the defect emission only (~1150 nm). The videos of defect blinking were recorded for 10 min with a frame rate of 2–10 frames-per-second.

**Covalent incorporation of defects into (6,5)-SWCNTs.** Aryl defects were chemically incorporated into the nanotube structure using a light-activated diazonium reaction[36]. Briefly, we conducted the chemistry by mixing aqueous suspensions of (6,5)-SWCNTs with 4-methoxybenzenediazonium tetrafluoroborate (Sigma Aldrich, 98%) or 4-nitrobenzenediazonium tetrafl) uroborate (synthesized from 4-nitroaniline and nitrous acid) then irradiating the solution with 565 nm light using a Nanolog spectrometer (Horiba Jobin Yvon). The reaction was monitored in situ by measuring the evolution of a new emission that originates from the created defect sites, which is red-shifted from the intrinsic nanotube emission. The reaction was terminated when the desired defect PL intensity was reached by diluting the solution to 3.2% wt/v sodium deoxycholate (DOC, Sigma Aldrich, 97%)-D$_2$O using 4% wt/v DOC-D$_2$O. For (6,5)-SWCNTs containing two types of defects (MeOAr- and NO$_2$Ar-) a two-step reaction was performed by first reacting (6,5)-SWCNTs with 4-methxoyaryl diazonium salt and then 4-nitroaryl one. We confirmed the successful incorporation of each defect type at each step by PL spectroscopy (Supplementary Fig. 6). The pristine (6,5)-SWCNT has a characteristic $E_{11}$ peak at ~988 nm. Reaction with 4-methxoyaryl diazonium created a new emission peak that was 155 meV lower in energy than the $E_{11}$ peak, confirming the successful incorporation of MeOAr-defects. Following the addition of the 4-nitroaryl diazonium reactant, the intensity of this defect-related new emission peak increased while its emission energy further red-shifted to 170 meV below the $E_{11}$ peak, as the NO$_2$Ar-defects become dominating the final product.

## Data availability
The datasets generated and/or analysed during the current study are available from the corresponding author on reasonable request.

## Code availability
The custom MATLAB codes developed in the current study are available from the corresponding author on reasonable request.

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

## Acknowledgements

M.K. acknowledges the Millard and Lee Alexander Fellowship in Chemistry from the University of Maryland. X.W. and H.Q. are partially supported by the Center for Enhanced Nanofluidic Transport (CENT), an Energy Frontier Research Center funded by the U.S. Department of Energy, Office of Science, Basic Energy Sciences under Award # DE-SC0019112 for studying the roles of defects in nanofluidic transport. This work also makes use of instrumentation components funded in part by NSF (Grant No. CHE-1507974, which is continued as CHE-1904488) and NIH/NIGMS (Grant No. R01GM114167). We thank D. A. Heller, P. Jena, and D. Roxbury for intriguing discussions and sharing a MATLAB code for correcting non-uniform laser excitation in hyperspectral data, which is adapted to produce Fig. 4c, and Photon Etc. and Xenics for helping with the instrument integration.

## Author contributions

Y.H.W. conceived the experiments and supervised the project; X.W., M.K. and H.Q. performed the measurements; X.W. analyzed the results; X.W. and Y.H.W. wrote the paper with inputs from all authors.
