## [Peer Review File · Nature Communications]

Reviewers' comments:

Reviewer #1 (Remarks to the Author):

The authors report a spectroscopic technique that allows single-defect imaging and analysis in the shortwave infrared (IR). Therein, high-resolution imaging with photoluminescent spectroscopic analysis is achieved by the use of an InGaAs detector cooled at -190 degrees C and of a fiduciary marker made of silicon windows transparent to the shortwave IR. In addition, the implementation of a read-while-integrate readout mode largely enhances the signal to noise ratio compared to a conventional integrate-then-read mode. Finally the authors applied the developed technique to the single-walled carbon nanotubes (SWCNTs) with chemically-incorporated defects and successfully observe several types of defect structures on the SWCNTs in the nanometer scale.

The well-assembled system for the defect analysis is useful and important to precisely characterize and understand nanoscale structures and functions not only for SWCNTs but also for other nanomaterials functionalized by defects. Therefore, after the revision regarding the following comments, this paper would be recommended for publication.

1) Based on the result shown in Figure 4, the authors suggest the existence of three types of defect sites and conclude that the site iii contains both of 4-nitroaryl and 4-methoxyaryl structures. However, the peak wavelengths of the 4-nitroaryl defect and the 4-methoxyaryl defect are not same between those of the site iii and the site i or the site ii although the authors have reported that the structure of aryl defect dominates the PL wavelength (ref. #24). The reason of the observed wavelength shifts between these sites should be discussed in the manuscript to support the authors' conclusion. In addition, discussion about why the three defects are located inhomogeneously is necessary. For example, the site i is formed particularly around the tube edge.

2) In order to understand which type of the defects of the analyzed SWCNTs is the major structure, it would be crucial to show the proportion of the defects having bimodal and multiple intensity steps together with their spatial distribution along the nanotubes.

3) On page 8, 2nd paragraph, the authors unexpectedly observe a defect coupling phenomenon. I agree that quantitative analysis of the coupling effect could be a future work, but the authors may be able to show the number or the proportion of such coupling defects per one nanotube, which would be helpful information for the researchers in the defect-engineering fields.

4) On page 5, line 2, the authors described “with this substrate design the collection efficiency of the nanotube PL is ~5-times higher than when imaged on bare glass”. The corresponding experimental data should appear in Supplementary Information for clarification.

5) The spatial resolution of the developed system needs to be described.

6) The relationship between the design of the silicon-based fiduciary marker (size and pitch) and the 5-nm drift correction precision needs to be explained. In other words, it should be elucidated what kinds of the design parameters influence the precision in this spectroscopy.

7) For better understanding of readers in various research fields, the definition of “shortwave infrared (with the exact wavelength region)” and “color centers” should be described in the manuscript.

Reviewer #2 (Remarks to the Author):

This manuscript reports a new design of high throughput single-defect spectroscopy in the shortwave infrared, which was demonstrated to be capable of quantitatively and spectrally resolving chemical defects on the SWCNT semiconductor hosts. Their clever design includes several aspects, including cooling the typical InGaAs detector array down to -190 °C, implementing a non-destructive readout scheme called RWI, and setting a fiduciary marker that works in the shortwave IR to achieve a correction precision as high as 5 nm. Based on the amazing two orders-of-magnitude improvement of the signal-to-noise ratio, this technique can quantify the number of defects within a diffraction limited spot and spatially resolve them with a resolution of 15 nm in the shortwave infrared with simultaneous hyperspectral capabilities to unambiguously determine their chemical identities. With the convincing data on resolving chemical defects in SWCNT semiconductors at the single defect limit, I recommend acceptance after minor revisions considering the following comments.

1. According to their description on the simple design of the Au/polystyrene substrate, the collection efficiency of the nanotube PL is ~5-times higher than when imaged on bare glass. I am wondering how such an improvement can be achieved?
2. Cooling the typical InGaAs detector array down to -190 °C seems to be key trick of their technique. They did mention that in other conventional spectroscopy systems, liquid-N₂ cooled InGaAs detectors are typically operated at temperatures no lower than -85 °C, but no detailed reason is explained. Would a further cooling to -190 °C cause other side-effects? Is special technique needed to reach such a low temperature? This should be clarified in more detailed in the Methods.

3. In Fig. 3c, except for SWCNTs and the markers labelled by arrows, there are many white spots. Are they other shorter SWCNTs?
4. Figures 3a and b are not clear for me. For instance, in Figure 3a, the blue spot is the typical PL image as I understood, why is there a red center? Figure 3b seem to be the zoomed image of the white box in Figure 3a, why are there dotted circles? According to the caption, the dotted circles correspond to localization precision, does this mean specific functional group (defect site)?
5. In Figure 4, a carefully designed functionalized (6,5)-SWCNTs containing two types of defects (MeOAr- and NO₂Ar-) were used. Although the synthesis details were described in Methods, characterization of this claimed structure is missing. I guess that this was reported previously, but a brief discussion is needed to prove that this sample is indeed what they expected.

Point-by-point response to reviewers' comments

Reviewer #1 (Remarks to the Author):

The authors report a spectroscopic technique that allows single-defect imaging and analysis in the shortwave infrared (IR). Therein, high-resolution imaging with photoluminescent spectroscopic analysis is achieved by the use of an InGaAs detector cooled at -190 degrees C and of a fiduciary marker made of silicon windows transparent to the shortwave IR. In addition, the implementation of a read-while-integrate readout mode largely enhances the signal to noise ratio compared to a conventional integrate-then-read mode. Finally the authors applied the developed technique to the single-walled carbon nanotubes (SWCNTs) with chemically-incorporated defects and successfully observe several types of defect structures on the SWCNTs in the nanometer scale.

The well-assembled system for the defect analysis is useful and important to precisely characterize and understand nanoscale structures and functions not only for SWCNTs but also for other nanomaterials functionalized by defects. Therefore, after the revision regarding the following comments, this paper would be recommended for publication.

Response: We thank the reviewer for the positive comments.

“1) Based on the result shown in Figure 4, the authors suggest the existence of three types of defect sites and conclude that the site iii contains both of 4-nitroaryl and 4-methoxyaryl structures. However, the peak wavelengths of the 4-nitroaryl defect and the 4-methoxyaryl defect are not same between those of the site iii and the site i or the site ii although the authors have reported that the structure of aryl defect dominates the PL wavelength (ref. #24). The reason of the observed wavelength shifts between these sites should be discussed in the manuscript to support the authors' conclusion. In addition, discussion about why the three defects are located inhomogeneously is necessary. For example, the site i is formed particularly around the tube edge.”

Response: The observed wavelength shifts at site iii could be attributed to the difference in the local dielectric environment (e.g., surfactant wrapping; Harvey et al. ACS Appl. Mater. Interfaces, **2017**, 9, 37947) and endohedral filling of water (Cambré et al. ACS Nano **2012**, 6, 3, 2649-2655) given the following factors: first, the 5~6 meV shift is significantly smaller than the energy difference between the two types of defects (~19 meV); second, the E₁₁ emission at site iii also shows a similar redshift in the emission energy (we now add Figure S5 to support it).

During the dispersion of the SWCNTs in the surfactant/water solution, the SWCNTs are wrapped in surfactant and the water molecule can spontaneously enter the cavity of the nanotubes through the two open ends (Cambré et al. ACS Nano **2012**, 6, 3, 2649-2655). When the SWCNTs are deposited and dried on the imaging substrate, the surfactant shell collapses, resulting in inhomogeneous coverage on the nanotube surface, along with partial evaporation of the water molecules inside the tube structure. The redshift we observed at site iii could be caused by variation in the local environment due to these factors. To clarify this point, we now include a brief discussion in the revised manuscript on page 9:

“We note that the two defects at site iii show a 5~6 meV redshift in emission energy compared to the corresponding defects at sites i and ii. This shift is relatively small compared to the energy difference between these two types of defects (~ 19 meV)²⁸ and was also observed in the E₁₁ emission (Figure S5). We attribute this behavior to the different local dielectric environment caused by inhomogeneous surfactant coverage on the nanotube surface³³ or the partial filling of water inside the SWCNT³⁴.”

Figure S5. The normalized PL spectra of a (6,5)-SWCNT with two types of defects, methoxyaryl and nitroaryl. The E₁₁ emission of at site iii has a 4 meV redshift from that of site ii.

Regarding the spatial distribution of the defects along the nanotubes, the reviewer has also raised an interesting question. However, the defect distribution is itself an interesting topic that requires a separate study. The current work is focused on the imaging technique rather than the defect chemistry. We have now added a note on page 8 in the manuscript: “Note that the defects are located inhomogeneously along the nanotube length.”

2) In order to understand which type of the defects of the analyzed SWCNTs is the major structure, it would be crucial to show the proportion of the defects having bimodal and multiple intensity steps together with their spatial distribution along the nanotubes.

Response: The intensity steps are not associated with the type of defect but with the number of defects at each site. We identified the defect types directly by their emission energy from the PL spectra.

3) On page 8, 2nd paragraph, the authors unexpectedly observe a defect coupling phenomenon. I agree that quantitative analysis of the coupling effect could be a future work, but the authors may be able to show the number or the proportion of such coupling defects per one nanotube, which would be helpful information for the researchers in the defect-engineering fields.

Response: We observed this coupling effect only occasionally (~5%). This probability should strongly depend on the density of defects and further studies are required to quantitatively measure the effect. We have added a note in the text on page 8.

“We note that this coupling effect was observed only occasionally (~5%) in the current study and will be further exploited in future experiments.”

4) On page 5, line 2, the authors described “with this substrate design the collection efficiency of the nanotube PL is ~5-times higher than when imaged on bare glass”. The corresponding experimental data should appear in Supplementary Information for clarification.

Response: The reflective gold layer acts as a mirror that enhances both the absorption of excitation photons by the carbon nanotube and the collection of emission photons by the objective. The insulating PS layer provides a neutral surface, which minimizes surface charges that would otherwise quench the nanotube PL. The above three factors collectively contributed to the ~5-times PL enhancement. To better convey the advantages of this imaging surface, we now incorporate the experimental data (as the new Figure S2) comparing the PL intensity of nanotubes deposited on Au/polystyrene and bare glass in the Supplementary Information.

Figure S2 | The polystyrene/gold (PS/Au) substrate increased the PL intensity by ca. 5-fold. PL images of SWCNTs on (a) bare glass and (b) the PS/Au substrate. The excitation wavelength was 730 nm and the integration time was 2 s. The color bar is the PL intensity ADU count. (c, d) The corresponding histograms of the SWCNT PL intensity distribution based on the PL images shown in (a) and (b).

We have also made a note in the main text of the revised manuscript (Page 5):

“This observed increase can be attributed to the reflective gold layer acting as a mirror to enhance both the absorption of excitation photons by the carbon nanotubes and the collection efficiency of emission photons by the objective, as well as the PS layer, which minimizes surface charges that would otherwise quench the PL.”

5) The spatial resolution of the developed system needs to be described.

Response: We have noted on pages 8 and 10 in the text that the spatial resolution is 15 nm.

6) The relationship between the design of the silicon-based fiduciary marker (size and pitch) and the 5-nm drift correction precision needs to be explained. In other words, it should be elucidated what kinds of the design parameters influence the precision in this spectroscopy.

Response: The size and pitch of the fiduciary markers themselves do not directly affect the drift correction precision as long as the markers are not overlapping with others. However, the size and pitch of the markers determine the number of markers that can be present in the field of view. Since the drift correction precision was calculated using the drift curves from all markers in an image, the more markers on the image the better correction precision that we can achieve. To achieve 5-nm drift correction precision, we ensured there were at least 5 markers in the field of view.

We added a note to provide this information on page 5 of the revised manuscript.

“We note that to achieve 5-nm drift correction precision, we ensured there were at least 5 markers in the field of view.”

7) For better understanding of readers in various research fields, the definition of “shortwave infrared (with the exact wavelength region)” and “color centers” should be described in the manuscript.

Response: We agree with the reviewer’s suggestion and now include the definition of “shortwave infrared” and “color centers” on page 2 of the revised manuscript.

“The shortwave infrared (typically 900 – 1700 nm, but may also refer to 1–3 μm) is a spectral window that presents exciting opportunities for bioimaging.”

“Color centers (emissive point defects in crystal lattices) that emit light in the shortwave IR are also important for non-invasive in vivo fluorescent imaging.”

Reviewer #2 (Remarks to the Author):

“This manuscript reports a new design of high throughput single-defect spectroscopy in the shortwave infrared, which was demonstrated to be capable of quantitatively and spectrally resolving chemical defects on the SWCNT semiconductor hosts. Their clever design includes several aspects, including cooling the typical InGaAs detector array down to -190 oC,

implementing a non-destructive readout scheme called RWI, and setting a fiduciary marker that works in the shortwave IR to achieve a correction precision as high as 5 nm. Based on the amazing two orders-of-magnitude improvement of the signal-to-noise ratio, this technique can quantify the number of defects within a diffraction limited spot and spatially resolve them with a resolution of 15 nm in the shortwave infrared with simultaneous hyperspectral capabilities to unambiguously determine their chemical identities. With the convincing data on resolving chemical defects in SWCNT semiconductors at the single defect limit, I recommend acceptance after minor revisions considering the following comments.

1. According to their description on the simple design of the Au/polystyrene substrate, the collection efficiency of the nanotube PL is ~5-times higher than when imaged on bare glass. I am wondering how such an improvement can be achieved?"

Response: The reflective gold layer acts as a mirror that can nearly double both the excitation photons absorbed by the carbon nanotube and the emission photons collected by the objective. The PS layer provides a neutral surface, which minimizes surface charges that would otherwise quench the nanotube PL. These three factors collectively contributed to the ~5-times PL enhancement.

To better convey the advantages of this imaging surface, we have now incorporated the experimental data (as the new Figure S2) comparing the PL intensity of nanotubes deposited on Au/polystyrene and bare glass in the Supplementary Information.

Figure S2 | The polystyrene/gold (PS/Au) substrate increased the SWCNT PL signal by ca. 5-fold. PL images of SWCNTs on (a) bare glass and (b) the PS/Au substrate. The excitation wavelength was 730 nm and the integration time was 2 s. The color bar is the PL intensity ADU count. (c, d) The corresponding histograms of the SWCNT PL intensity distribution based on the PL images shown in (a) and (b).

We have also added a note about this observation in the main text of the revised manuscript (Page 5):

“This observed increase can be attributed to the reflective gold layer acting as a mirror to enhance both the absorption of excitation photons by the carbon nanotubes and the collection efficiency of emission photons by the objective, as well as the PS layer, which minimizes surface charges that would otherwise quench the PL.”

“2. Cooling the typical InGaAs detector array down to -190 oC seems to be key trick of their technique. They did mention that in other conventional spectroscopy systems, liquid-N2 cooled InGaAs detectors are typically operated at temperatures no lower than -85 °C, but no detailed reason is explained. Would a further cooling to -190 oC cause other side-effects? Is special technique needed to reach such a low tempertature? This should be clarified in more detailed in the Methods.”

Response: Before answering the reviewer’s comments, we’d like to correct our initial statement that liquid-N2 cooled InGaAs detectors are typically operated at temperatures no lower than -85 °C. Some liquid-N2 cooled InGaAs detectors can be operated at a temperature slightly lower than -85 °C, e.g. -100 °C for 2D-OMA V from Princeton Instrument. We have now changed the statement on page 5 to: “Note that liquid-N2 cooled InGaAs detectors are typically operated at temperatures no lower than -100 °C.”

We note that both cooling the detector array to nearly the true liquid-nitrogen temperature and implementing a non-destructive readout scheme contribute to the high signal-to-noise ratio achieved in our system. Due to the lower bandgap and crystal quality, InGaAs cameras have much higher dark current than Si-CCD cameras. The typical dark current for an InGaAs camera is on the level of 5000 e⁻/pixel/sec at approximately -100 °C (e.g. 2D-OMA V, Princeton Instrument), compared to 0.002 e⁻/pixel/sec for a Si-CCD camera at -70 °C. For this reason, there is little benefit to cool Si-CCD below -85°C, however, for InGaAs detectors deep cooling provides a solution to minimize the dark current thereby significantly improving the signal to noise ratio as we demonstrated here. The detector dark current reduces by a factor of 2 with every reduction of 8 °C of the detector temperature, lowering the dark current to merely 10 e⁻/pixel/sec at a detector temperature of -190 °C.

A side effect from this deep cooling is that the detection range on the long-wavelength side is reduced by 8 nm for every 10 °C of cooling. Further cooling the detector from 170 K to 80 K reduces the upper bound of the detection range from 1.62 μm (170 K) to 1.55 μm (80 K).

Cooling the detector to -190 °C and maintaining it at that low temperature, which is significantly lower than -100 °C, also require significant improvement in dewar efficiency and device design to accommodate the large thermal stress as the sensor goes through the cooling-heating cycles. This is another major reason that most detectors are not cooled below -100 °C. Rather, the rate of liquid-N₂ evaporation is controlled such that the sensor temperature is maintained at above -100 °C.

“3. In Fig. 2c, except for SWCNTs and the markers labelled by arrows, there are many white spots. Are they other shorter SWCNTs?”

Response: Yes. They are shorter SWCNTs. The length distribution of commercial SWCNTs (CoMoCat SG65i in this case) can range from tens of nanometers to several microns. We have clarified this in the caption of Fig. 2c: “Nanotubes with length smaller than the diffraction limit (~600 nm for a NA 0.85 objective collecting emission at 1000 nm) appear as white dots in the image.”

“4. Figures 3a and b are not clear for me. For instance, in Figure 3a, the blue spot is the typical PL image as I understood, why is there a red center? Figure 3b seem to be the zoomed image of the white box in Figure 3a, why are there dotted circles? According to the caption, the dotted circles correspond to localization precision, does this mean specific functional group (defect site)?”

Response: The red centers in Figure 3a represent the defect locations obtained from our super-resolution technique. The dotted circles in Figure 3b are the localization precision of this method. We calculated these values using the localization theory (ref. 31):

$$\text{Var}_x = \frac{S_a^2}{N} \times \left(\frac{16}{9} + \frac{8\pi S_a^2 b^2}{Na^2} \right)$$

Var_x	The variance of the localization position in the X dimension when fitting a Gaussian 2D function to the emitter intensity profile.
S_a	The standard deviation of the Gaussian function (s) adjusted for square pixels. The adjustment is computed as: $S_a = \sqrt{s^2 + a^2/12}$.
N	The number of photons in the localization.
b^2	The expected number of photons per pixel from a background.
a	The pixel size in nm.

“5. In Figure 4, a carefully designed functionalized (6,5)-SWCNTs containing two types of defects (MeOAr- and NO₂Ar-) were used. Although the synthesis details were described in Methods, characterization of this claimed structure is missing. I guess that this was reported previously, but a brief discussion is needed to prove that this sample is indeed what they expected.”

Response: We thank the reviewer for the suggestion and have now include a brief discussion on page 11 in the revised manuscript to make this information clear.

“For (6,5)-SWCNTs containing two types of defects (methoxyaryl (MeOAr-) and nitroaryl (NO₂Ar-)) a two-step reaction was performed by first reacting (6,5)-SWCNTs with 4-methoxyaryl diazonium salt and then 4-nitroaryl one. We confirmed the successful incorporation of each defect type at each step by PL spectroscopy (Figure S6). The pristine (6,5)-SWCNT has a characteristic E₁₁ peak at ~988 nm. Reaction with 4-methoxyaryl diazonium created a new emission peak that was 155 meV lower in energy than the E₁₁ peak, confirming the successful

incorporation of MeOAr-defects. Following the addition of the 4-nitroaryl diazonium reactant, the intensity of this defect-related new emission peak increased while its emission energy further red-shifted to 170 meV below the E_{11} peak, as the NO_2Ar -defects become dominating the final product.”

Figure S6 | Incorporating two types of defects in the same nanotube. The ensemble level PL spectra of the pristine (6,5)-SWCNT sample (black), reacting first with 4-methoxyaryl diazonium (red), then with 4-nitroaryl diazonium (blue), and only with 4-nitroaryl diazonium (green). The excitation wavelength is 565 nm and the integration time is 3 s.

REVIEWERS' COMMENTS:

Reviewer #1 (Remarks to the Author):

The authors report a single-defect imaging technique with spectral analyses in the shortwave IR. Most of authors' revision satisfied my comments and questions. Regarding the response for my comment #2, it would be better if the authors can make a comment about which is the major component, a single defect showing a bimodal intensity change or a clustered defect showing multiple intensity steps. However, this paper can be accepted in the current form.

Reviewer #2 (Remarks to the Author):

The authors had addressed my comments well, thus it can be published in the present form.

Point-by-point response to reviewers' comments

Reviewer #1 (Remarks to the Author):

The authors report a single-defect imaging technique with spectral analyses in the shortwave IR. Most of authors' revision satisfied my comments and questions. Regarding the response for my comment #2, it would be better if the authors can make a comment about which is the major component, a single defect showing a bimodal intensity change or a clustered defect showing multiple intensity steps. However, this paper can be accepted in the current form.

Response: Clustered defects showing multiple intensity steps are the major component. We have now added a note on page 8 of the revised manuscript: “We note that the majority (>80%) of defect emitting sites observed in the current study show multiple intensity steps, indicating the prevalence of clustered defects.”

Reviewer #2 (Remarks to the Author):

The authors had addressed my comments well, thus it can be published in the present form.

Response: We again thank the reviewer for the constructive comments.